# Multiple Testing, Cut-Point Optimization, and Signs of Publication Bias in Prognostic FDG–PET Imaging Studies of Head and Neck and Lung Cancer: A Review and Meta-Analysis

**DOI:** 10.3390/diagnostics10121030

**Published:** 2020-12-01

**Authors:** Malene M. Clausen, Ivan R. Vogelius, Andreas Kjær, Søren M. Bentzen

**Affiliations:** 1Department of Clinical Physiology, Nuclear Medicine & PET and Cluster for Molecular Imaging, Copenhagen University Hospital, Rigshospitalet and University of Copenhagen, 2100 Copenhagen, Denmark; akjaer@sund.ku.dk; 2Department of Oncology, Copenhagen University Hospital, Rigshospitalet, 2100 Copenhagen, Denmark; ivan.richter.vogelius@regionh.dk; 3Division of Biostatistics and Bioinformatics, University of Maryland School of Medicine, Baltimore, MD 21201, USA; sbentzen@som.umaryland.edu

**Keywords:** FDG–PET/CT, HNSCC, NSCLC, prognostication, publication bias

## Abstract

Positron emission tomography (PET) imaging with 2-deoxy-2-[^18^F]-fluorodeoxyglucose (FDG) was proposed as prognostic marker in radiotherapy. Various uptake metrics and cut points were used, potentially leading to inflated effect estimates. Here, we performed a meta-analysis and systematic review of the prognostic value of pretreatment FDG–PET in head and neck squamous cell carcinoma (HNSCC) and non-small cell lung cancer (NSCLC), with tests for publication bias. Hazard ratio (HR) for overall survival (OS), disease free survival (DFS), and local control was extracted or derived from the 57 studies included. Test for publication bias was performed, and the number of statistical tests and cut-point optimizations were registered. Eggers regression related to correlation of SUVmax with OS/DFS yielded *p* = 0.08/*p* = 0.02 for HNSCC and *p* < 0.001/*p* = 0.014 for NSCLC. No outcomes showed significant correlation with SUVmax, when adjusting for publication bias effect, whereas all four showed a correlation in the conventional meta-analysis. The number of statistical tests and cut points were high with no indication of improvement over time. Our analysis showed significant evidence of publication bias leading to inflated estimates of the prognostic value of SUVmax. We suggest that improved management of these complexities, including predefined statistical analysis plans, are critical for a reliable assessment of FDG–PET.

## 1. Introduction

Positron emission tomography (PET) offers a non-invasive method to assess functional biological characteristics of a tumor in an individual patient with cancer. A number of positron emitting tracers were developed to study various aspects of tumor biology [1,2,3,4,5,6,7]. However, clinical practice in cancer PET imaging is still dominated by very few tracers, with 2-deoxy-2-[^18^F]-fluorodeoxyglucose (FDG) being the clinical workhorse for most tumor sites. FDG–PET imaging is primarily used for staging purposes, as a supplement to anatomical images, but advances in the availability of PET imaging led to an increased interest in the feasibility of PET guided radiotherapy planning [8]. Several studies investigated the prognostic value of FDG–PET, and dose escalation to PET-positive areas within the tumor is one of the potential strategies for increasing effect of radiotherapy [9,10,11,12].

Despite the widespread use of FDG, only few prospective studies exist for this tracer—the value of FDG as prognostic factor was mainly tested in retrospective cohort studies, for example [13]. There are major limitations with this approach of evidence-generation in medicine. For example, the lack of prospective clinical studies with study registration, formalized sample size estimation and a predefined statistical analysis plans. All of this leads to a risk of “fishing expeditions” with a high risk of false positive findings, exaggerated effect sizes, and subsequent publication bias. Multiple testing is a well-described contributor to false positive findings; when comparisons are made for several subgroups or multiple variables, without adjustment of the type I error rate (false positives).

Searching for a standardized uptake value (SUV) cut-point for dichotomization of the patient group into a poor vs. good prognosis group, which minimizes the *p*-value when comparing outcome in the resulting groups, is an approach frequently used in imaging studies. This “optimization” approach invalidates a simple interpretation of the resulting *p*-value and is associated with a substantial increase in the rate of type I errors [14,15]. Deciding before the start of analyses to use the median SUV as a cut-point, is unbiased. However, in many cases this might not be a biologically meaningful way to classify patients into prognostic subgroups.

In the current study, we reviewed the methodology of published studies of the prognostic value of FDG in head and neck squamous cell carcinoma (HNSCC) and non-small cell lung cancer (NSCLC), and assessed the evidence of publication bias. We included studies where patients received radiotherapy (RT), possibly in combination with other treatment modalities. The Hazard Ratio (HR) for overall survival (OS), disease free survival (DFS), or local control (LC) were considered as outcomes, and analyzed as a function of FDG uptake metrics.

## 2. Materials and Methods

### 2.1. Search Strategy and Eligibility Criteria for Studies

Published reports on the prognostic value of pretreatment FDG–PET in two common tumors, HNSCC and NSCLC, were included.

We searched PubMed for published reports using the following search strings with ‘human’ filter:


*HNSCC: (FDG OR “18-F”) AND (“Head and neck” OR “HNSCC” OR “SCCHN”) AND (radiotherapy OR chemoradi* OR radio* OR chemo-radi*) NOT review*



*NSCLC: (FDG OR “18-F”) AND (“non-small cell lung cancer” OR “NSCLC”) AND (radiotherapy OR chemoradi* OR radio* OR chemo-radi*) NOT review*


In addition, manual screening of the selected articles, reviews, and meta-analyses were used to complement the search. The final date of the search was 1 November 2018, and we did not restrict publication date prior to this date. In this analysis, only articles in English were included. Studies reporting the HR for OS, DFS or LC, versus SUVmax or SUVpeak, were included. The primary treatment was required to include RT, but studies with mixed cohorts including RT, chemoradiotherapy, and surgery were also allowed. However, studies with only few patients receiving RT were excluded [16]. No restrictions on the study design were used. Where multiple reports with overlapping patient cohorts were available, only data from the largest study were included.

### 2.2. Data Extraction

Data were extracted for each study by MMC and entered into the meta-analysis software Review Manager (RevMan) software version 5.3 [17]. There was no attempt to obtain unpublished data. Data were analyzed by MMC and IRV. The HR for OS, DFS, and LC for each trial was extracted or derived from the available data. An HR above 1 implied a survival benefit for lower SUVmax. Both the HR and its confidence interval (CI) were required from each study for inclusion in the meta-analysis. Data from multivariate (MVA) were prioritized over univariate (UVA) analysis, when both were reported.

If the HR with CI was not stated in a report, one of two methods were used for its estimation. If the HR was given with a *p*-value, but without the CI, we assumed a normal distribution of the logarithm of the HR and estimated the CI by first finding the z-parameter of the normal distribution pertaining to the reported *p*-value. We then calculated the standard error of the ln(HR) estimate as $S_{\ln\left(\mathrm{HR}\right)}=\frac{\ln\left(\mathrm{HR}\right)}{z}$.

A number of studies did not report an HR, but only a *p*-value, together with the outcome at one or more specified points in time, in most cases in the form of a plot of Kaplan–Meier curves. In these cases, we estimated the HR from the relationship $R\left(t\right)=\frac{\ln\left(p_{2}\left(t\right)\right)}{\ln\left(p_{1}\left(t\right)\right)}$, where p_1_ is the Kaplan–Meier estimate for low SUVmax at time t and p_2_ is the estimate for high SUVmax. When possible, we sampled the ratio at multiple time-points, ranging from the first time an event occurred in both groups to the end of follow-up, and averaged the resulting HR(t) estimates. The CI was then calculated from the *p*-value as explained above. The methodology was previously described in more detail [18]. Variances were then calculated and used as study weights in meta-analysis, using RevMan [17].

Assessment of publication bias was made visually by ordering the studies in forest plots according to variance and by funnel plots. We did not systematically assess the risk of bias in the individual studies. The risk of bias across studies was assessed using the so-called *Egger–Var* method as a formal test for publication bias [19]. We performed quantitative assessment using the Egger’s method as follows. For each endpoint and comparison, a linear fit of log HR versus standard error of the study estimate was performed to assess if the included studies effect size depended on study precision. Formally, the regression equation was
(1)$$\ln\left({\mathrm{HR}}_{i}\right)=\alpha +{\beta *\mathrm{SE}}_{i}+\varepsilon_{i},$$
weighted by the inverse variance of each study. Here, α and β were the fitting parameters, SE_i_ was the standard error of lnHR of the ith study, and ε was the residual error assumed to have a normal distribution. If β was different from zero at the 95% confidence level, it was concluded that the effect size estimates depended on the study precision—a clear indication of publication bias. α in Equation (1) was the extrapolation to zero SE and was used to estimate a publication bias corrected value of ln(HR).

Turning to the assessment of the number of statistical tests and the use of cut-point optimization, the assessment was made independently by MMC and IRV and disagreements were resolved by a consensus meeting. Only statistical tests related to the association of an imaging metric on one side and oncological outcome or baseline characteristics on the other, were counted. When the same question was addressed in univariate and multivariate analysis, the corresponding *p*-value was only counted once. Similarly, it was only counted once if it was part of a rational model building procedure, including forward or backwards elimination. In cases where a large number of multivariate models with a functional image metric were examined, outside of a model building procedure, the multivariate tests were included in the assessment of a number of statistical tests that were investigated, e.g., Schwartz et al. [20].

## 3. Results

The study selection process for this analysis is presented in Figure 1. Of the 930 studies identified by the initial search, 57 were analyzable. A total of 178 full text articles were screened, and 133 of these were excluded, as data were not assessable (not reporting HR, univariate log-rank test without *p*-value, no cut-point for SUV). In other words, 25.3% of the screened full text reports were included. Twelve studies were added from manual cross-referencing of articles, reviews, and browsing for a total of 57 included studies; 27 studies in patients with HNSCC [20,21,22,23,24,25,26,27,28,29,30,31,32,33,34,35,36,37,38,39,40,41,42,43,44,45,46], and 30 studies in NSCLC [47,48,49,50,51,52,53,54,55,56,57,58,59,60,61,62,63,64,65,66,67,68,69,70,71,72,73,74,75,76].

Study characteristics are summarized in Table 1 and Table 2 for HNSCC and NSCLC, respectively. The vast majority of studies were retrospective analyses—20 studies of HNSCC (74%) and 26 studies of NSCLC (87%). The included studies comprised a total of 5102 patients—1704 patients in the HNSCC group and 3398 in the NSCLC group. The median study size was 74. The NSCLC studies were generally larger, with a median study sample size of 95 patients compared to 58 in the HNSCC group. Twenty-seven studies did not perform MVA for DFS or OS, which were the primary endpoints of this analysis. A few studies reported no events until long follow-up, which gave rise to additional uncertainty in the HR estimate [54,68]. One study reported no events in the low-uptake group, giving rise to an infinite HR estimate, and the study had to be excluded [77]. A single study was excluded due to problems with interpretation of the KM plots [78].

The patient cohorts in both the HNSCC and NSCLC group were quite heterogeneous with respect to stage, treatment, and follow-up time.

Figure 2 and Figure 3 display the forest plots for SUVmax as a predictor of DFS and OS for HNSCC and NSCLC, respectively, with the studies ordered according to inverse variance from top to bottom. There was a trend for HR to decrease with decreasing variance, and publication bias is therefore suspected. The HR estimate from the pooled analysis is shown by the diamond-shaped mark, and it favored low SUV. However, this should be interpreted with caution due to the suspicion of publication bias. The same trend was observed for LC (Appendix A). The data entered in RevMan and HR for UVA and MVA for all studies are listed in the Appendix A.

When Eggers regression was applied, neither OS nor DFS appeared to be significantly associated with FDG uptake in neither HNSCC or NSCLC. The regression slopes were significantly greater than zero in three of the four cases: DFS for HNSCC (*p* = 0.02) and both OS (*p* < 0.001) and DFS (*p* = 0.014) for NSCLC. See the supplement for details and the associated plots (Appendix A).

Figure 4 shows a plot of the number of patients, number of statistical tests, and number of cut-point optimizations against the year of publication. Unfortunately, there is little sign of a consistent improvement in study characteristics, i.e., number of statistical tests, number of cut-point analyses, and size of study population over time. For HNSCC studies, there was a statistically significant increase in the number of cut-point optimizations versus time (Spearman rho = 0.5, *p* = 0.02), while there was no significant increase in the number of patients for later studies and data were also in accordance with no change in the number of tests performed (Figure 4A). Data for the NSCLC studies are shown in Figure 4B, where the Spearman rank correlation coefficients were statistically in agreement with no change over time, in either the number of patients, number of cut-points, or number of tests performed (*p* > 0.27 for all coefficients).

Power calculations were not performed in any of the included studies, and only three studies mention or adjust for multiple testing [27,33,60].

## 4. Discussion

Publication bias is a well-known problem, enriching the literature with false/true positive studies that will not be balanced by other studies with negative findings that are more likely to remain unpublished. This in turn will inflate the effect size estimated from an intervention or the discriminatory power of a diagnostic test. The inflated effect sizes from individual studies will carry over to a meta-analysis [79], thus reducing the value of the meta-analysis in evidence-based medicine. Indeed, our systematic analysis of prognostic studies of FDG uptake found statistically significant evidence of publication bias. Small studies are at particular risk of inflated effect-size bias [80] and it is thus a concern that the median study size was only 58 and 95 patients in published HNSCC and NSCLC studies, respectively. The TRIPOD reporting guidelines [81], attempts to address the problem by requiring a sample size justification in reporting, but this is not provided in any of the studies included here. Additionally, it might be argued that the general reporting guidelines of TRIPOD, albeit relevant, are not sufficiently specific for adequate reporting of image-based prognostic studies. In particular, it is an important concern that a large number of possible predictors can be extracted from a PET scan—SUVmax, SUVpeak, SUVmean, MTV, and TLG, just to name a few. Multiple comparisons, post-hoc search for positive associations and scanning for ‘optimal’ cut-off values separating the low and the high uptake groups, increase the risk of false positive findings [82], as also discussed by Vesselle et al., in the context of FDG prognostication [73].

A limitation of our study was that we did not have access to individual patient data, which led to the exclusion of some reports. Most of the included studies were conducted as retrospective studies (80.7%), without a predefined data analysis plan. While this might be defendable in the explorative setting, it increases the risk of overestimating the effect size if the cohort studies are not followed by controlled trials or studies with pre-specified protocols. In particular, with FDG, we would argue that we are beyond the exploratory phase and should perform larger studies with predefined protocols, to unequivocally reveal the prognostic or predictive role of FDG uptake in cancers that are common in the two sites studied in the present work. With the high number of correlations that are testable in image-based prognostication, it appears prudent to require predefined research protocols and, perhaps, publication of raw data to allow independent validation of findings, regardless of the chosen cut-point or predictor. It is possible that a functional imaging specific extension to the TRIPOD or REMARK guidelines could be of use. When published studies perform tens of comparisons and multiple cut-point optimizations in datasets of less than 100 patients, and without correction for multiple comparisons, the field is bound to be dominated by false or exaggerated correlations, which will ultimately harm patients if applied in clinical decision making and harm a promising field of research by misusing resources.

It is a substantial challenge to accommodate cross-study synthesis of data in meta-analysis at the same time as allowing the individual authors to appropriately handle the coding of image metrics in their study. Decisions to use continuous coding of SUV, logarithmic transformation, or a limited number of cut-points are all fair (if performed correctly), but hampers the ability to perform a meta-analysis. It appears to us that the complexity of these analyses implies that publication of the raw modeling data is a necessity for meaningful synthesis of data. We believe that the observations of the current study imply that such a synthesis is necessary for real progress.

## 5. Conclusions

Functional imaging with FDG or other tracers remains a promising tool for prognostication, prediction, and treatment selection for cancer patients. However, the current study points to issues limiting the interpretation, including inadequate sample sizes, lack of predefined analysis plans, lack of correction for multiple testing, and post-hoc cut-point optimizations. These issues result in a high risk of inflated effect sizes or false positive correlations that must be addressed to avoid leading the field astray.

## Figures and Tables

**Figure 1 diagnostics-10-01030-f001:**
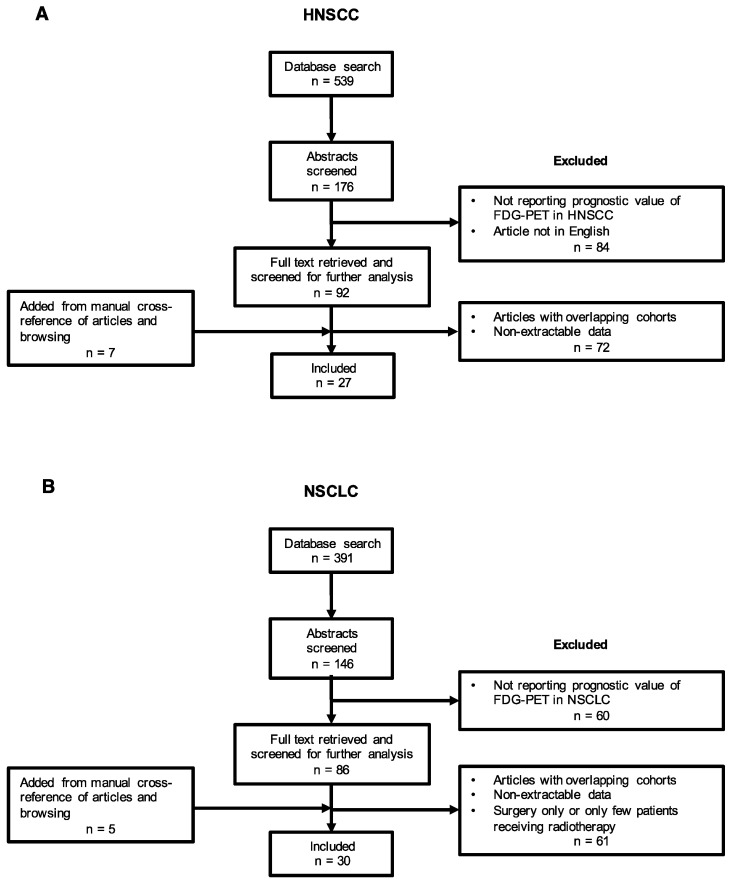
Flow diagram of the study selection process for the meta-analysis (**A**: HNSCC, **B**: NSCLC).

**Figure 2 diagnostics-10-01030-f002:**
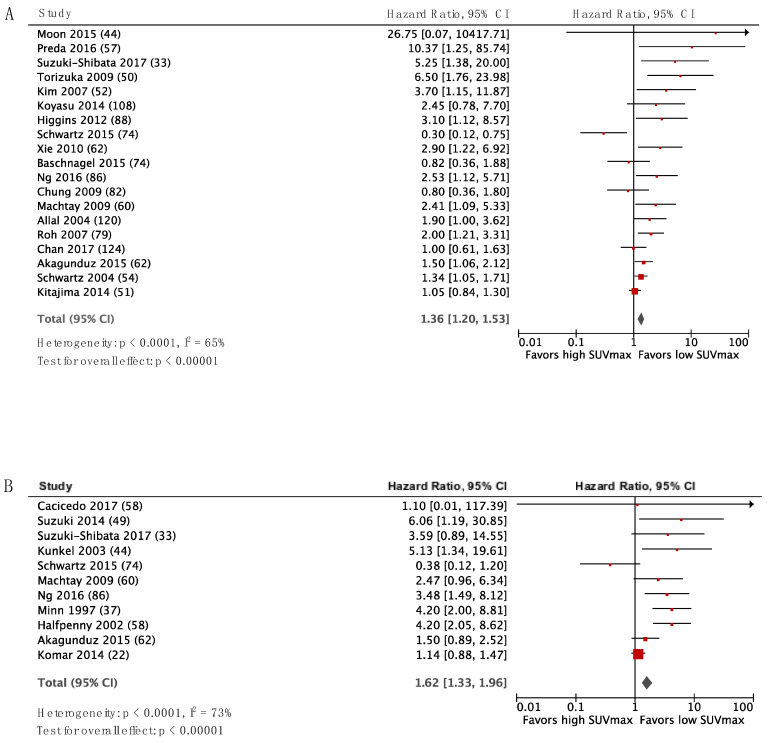
Forest plots for SUVmax as a predictor of DFS (**A**) and OS (**B**) in HNSCC. The number of patients included in each study is specified in parenthesis.

**Figure 3 diagnostics-10-01030-f003:**
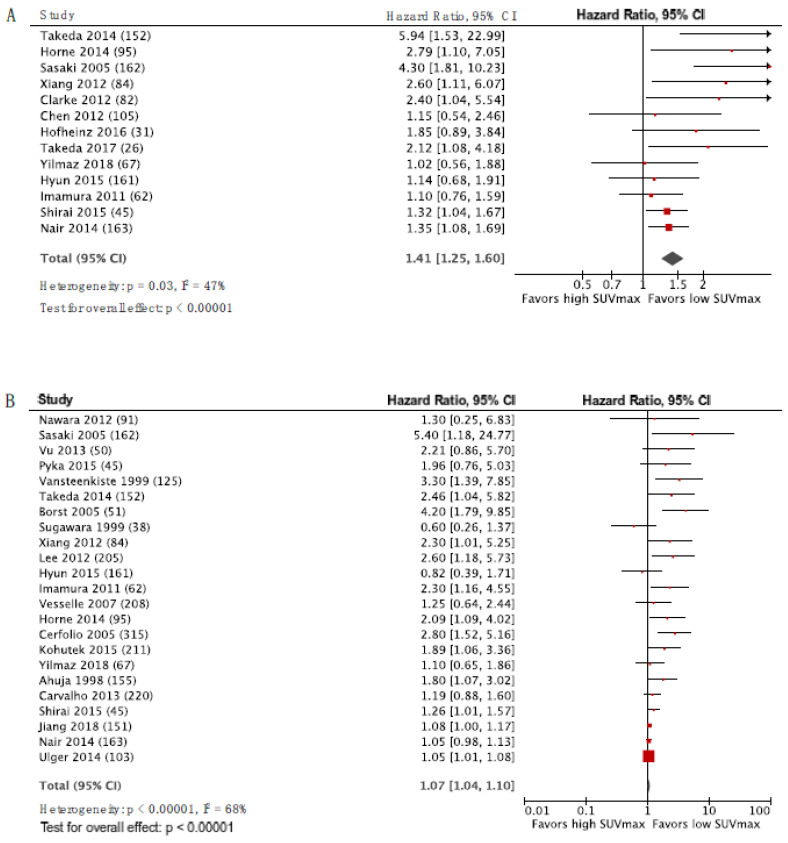
Forest plots for SUVmax as a predictor of DFS (**A**) and OS (**B**) in NSCLC. The number of patients included in each study is specified in parenthesis.

**Figure 4 diagnostics-10-01030-f004:**
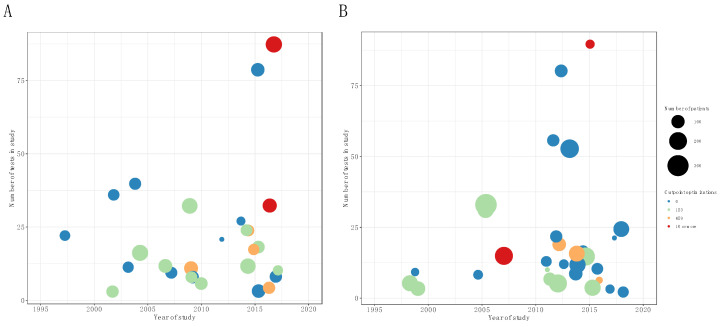
Bubble plot of number of patients included in study versus year of publication for HNSCC (**A**) and NSCLC (**B**). Bubble size is proportional to number of statistical tests counted in paper (large is problematic) and color denotes the number of cut-point optimization performed (the fewer the better).

**Table 1 diagnostics-10-01030-t001:** Characteristics of HNSCC studies included in the meta-analysis.

Author	Year	Tumor Type	Patients	Endpoints	MVA *	Uptake Metric	Cut-Off Value	SUV Threshold	Reconstruction Algorithm	Treatment	Stage	Median Follow Up Time	Data Extraction
Akagunduz et al.	2015	HN	62	LRFS, DFS, OS	No	SUVmax, SULmax, MTV	10.15 (SUL)	Fitted	-	RT/CRT		18 months	KM
*Allal* et al.	2004	HNSCC	120	LC, DFS, OS	Yes	SUVmax	4.76 3.5	Median Fitted	-	RT +/− CT, surgery +/− RT	I-IV	48 months	HR
Baschnagel et al.	2015	HNSCC	74	LC, LRC, DFS	Yes	SUVmax	13.8	Median	-	CRT	T1-T4, N0-N3	35 months	HR
*Brun* et al.	2002	HNSCC	47	CR, LRC, OS	No	SUV, MR	9.0	Median	Iterative ML	RT, CRT	II-IV	3.3 years	HR
*Cacicedo* et al.	2017	HNSCC	58	DFS, LRC, DMFS, OS	Yes	SUVmax	11.85 (SUV-T), 5.4 (SUV-N)	Median	-	Surgery + RT, RT +/− CT	III-IVB	31 months	KM
Chan et al.	2017	OHSCC	124	OS, RFS	Yes	SUVmax, SUVmean, MTV, TLG, entropy, contrast, busyness, complexity	14.22		OSEM	CRT	III-IV	28.7 months	HR
Chung et al.	2009	SCC	82	CR, DFS	Yes	MTV, SUV > 2.5	10.0	Median	OSEM	RT, CRT	I-IV	34.8 months	HR
Halfpenny et al.	2002	HNSCC	58	Survival	Yes	SUVpeak	10.0	Fitted	FBP	Surgery, +/−RT	I-IV	39 months	HR
Higgins et al.	2012	HNSCC	8	DFS, LRC, DMFS, OS	No	SUVmax, SUV mean, TLG	15.4	Median	OSEM	RT, CRT	III-IV (97%)	15 months	KM
Kim et al.	2007	OSCC	52	LC, DFS, OS	Yes	SUVmax	6.0	Median	-	Surgery +/− RT/CRT	I-IV	36 months	HR
Kitajima et al.	2014	Laryngeal	51	PFS, LC, NPFS, DMFS	Yes	SUVmax	4.6	Fitted	RAMLA	RT +/− CT, surgery +/− CRT		48.6 months	KM
Komar et al.	2014	HNSCC	22	OS	No	SUVmax, MATV	11.74	Median	-	Surgery +/− CRT, RT	I-IV	41 months	KM
Koyasu et al.	2014	SCC	108	DFS	Yes	SUVmax, MTV, TLG	10.0	Fitted	3D iterative	RT +/− CT, surgery +/− RT	I-IV	36.4 months	HR
Kunkel et al.	2003	OSCC	44	OS	Yes	SUVpeak	5.6	Median	-	RT (preop.) + surgery	I-IV	38 months	HR
Liao et al.	2009	OSCC	109	LC, DFS, DSS, OS	No	SUVmax	19.3	Median	ML, OSEM	Surgery + RT/CRT	III-IV	39 months	HR
Machtay et al.	2009	HNSCC	60	DFS, OS	Yes	SUVmax	9.0	Literature	-	RT, CRT, surgery + CRT/RT	I-IV	-	HR
*Minn* et al.	1997	HNSCC	37	OS	No	SUVlean, MR	9.0	Median	-	RT +/− surgery	II-IV	43 months	HR
Moon et al.	2015	NPC	44	DFS	Yes	SUVmax, SUVmean, TLG, MTV	7.8	Fitted	OSEM	CRT	II-IVB	40 months	HR
*Ng* et al.	2016	OHSCC	86	PFS, OS	Yes	SUVmax, SUVmean, TLG, MTV	19.44	Fitted	-	CRT	III-IVB	28 months	HR
Preda et al.	2016	HNSCC	57	DFS	Yes	SUVmax	5.75	Fitted	OSEM	Surgery + RT +/− CT, RT + CT	T1-T4, N0-N2	21.3 months	HR
Roh et al.	2007	SCC	79	DFS, LC, OS	No	SUVmax	8.0	Fitted	-	Surgery +/− RT or RT +/− CT	III-IV	36 months	KM
*Schwartz* et al.	2004	HNSCC	54	LRFS, DFS, OS	No	SUVmax	9.0	Median	FBP	RT +/− CT, surgery +/− RT	I-IV	17.5 months	KM
*Schwartz* et al.	2015	HNSCC	74	PFS, OS	No	SUVmax, MTV	15.07	Median	-	CRT	III-IV	4.2 years	HR
Suzuki et al.	2014	OPSCC + HPSCC	49	OS	Yes	SUVmax	8.0	Fitted	OSEM	Surgery + RT +/− CT, RT + CT	I-IV	33 months	HR
Suzuki-Shibata et al.	2017	OTSCC	33	PFS, OS	Yes	SUVmax, MTV	15.7	Fitted	FORE-OSEM	CRT	II-IVA	36 months	HR
Torizuka et al.	2009	HNSCC	50	LC, DFS	No	SUVpeak, SUV cont. variable	7.0	Fitted	OSEM	RT, CRT, surgery +/− CRT	I-IV	15 months	KM
Xie et al.	2010	NPC	62	OS, DFS	No	SUVmax	8.0	Fitted	-	CRT	III-IVB	61 months	KM

* Performed multivariate analysis (MVA) in regards to the endpoints analyzed in this study: DFS and OS. Studies listed with author in *italic* are performed as prospective studies. Other studies are retrospective studies. HNSCC: head and neck squamous cell carcinoma, HN: head and neck cancer, LRFS: local recurrence free survival, DFS: disease free survival, OS: overall survival, SUV: standardized uptake value, SUL: lean body mass corrected standardized uptake value, MTV: metabolic tumor volume, RT: radiotherapy, CRT: chemoradiotherapy, KM: Kaplan-Meier, LC: local control, HR: hazard ratio, LRC: locoregional control, CR: complete response, ML: maximum likelihood, DMFS: distant metastasis free survival, CT: chemotherapy, OHSCC: oropharyngeal or hypopharyngeal squamous cell carcinoma, RFS: recurrence free survival, TLG: total lesion glycolysis, OSEM: ordered-subset expectation maximum, SCC: squamous cell carcinoma, FBP: filtered back projection, OSCC: oral squamous cell carcinoma, PFS: progression free survival, NPFS: nodal progression free survival, RAMLA: row-action maximum-likelihood algorithm, MATV: metabolically active tumor volume, DSS: disease-specific survival, MR: metabolic rate, NPC: nasopharyngeal carcinoma, OPSCC: oropharyngeal squamous cell carcinoma, HPSCC: hypopharyngeal squamous cell carcinoma, OTSCC: oral tongue squamous cell carcinoma, and FORE-OSEM: Fourier rebinning- ordered-subset expectation maximum.

**Table 2 diagnostics-10-01030-t002:** Characteristics of NSCLC studies included in the meta-analysis.

Author	Year	Tumor Type	Patients	Endpoints	MVA *	Uptake Metric	Cut-Off Value	SUV Threshold	Reconstruction Algorithm	Treatment	Stage	Median Follow Up Time	Data Extraction
Ahuja et al.	1998	NSCLC	155	OS	Yes	SUVpeak (SUR) 80% of max	10.0	Fitted	-	Surgery, RT, CRT	I-IV	20.9 months	HR
Aoki et al.	2016	NSCLC	74	LC	Yes	SUVmax, AID	4.0	Literature	-	SBRT	I	24.5 months	HR
Borst et al.	2005	NSCLC	51	DSS, OS	No	SUVmax, SUV cont. variable	15.0	Median	OSEM	CRT	I-III	17 months	KM
*Carvalho* et al.	2013	NSCLC	220	OS	No	MTV, SUVmax, SUV	10.12	Median	OSEM	RT, CRT	I-IIIB	1.47 years	KM
Cerfolio et al.	2005	NSCLC	315	OS, DFS	Yes	SUVmax	10.0	Median	Iterative	Surgery +/− CRT	I-IV	26 months	HR
Chen et al.	2012	NSCLC	105	PFS, OS	Yes	TLG, MTV, SUVmax	15.0	Fitted	OSEM	Surgery, CT, RT or CRT	I-IV	3.1 years	HR
Clarke et al.	2012	NSCLC	82	OS, RFS, DFS, CSS, RR, LR, DM	No	SUVmax	4.75	Median	-	SBRT	I	2 years	KM
Hamamoto et al.	2011	NSCLC	26	LFF	No	SUVmax	5.0	Fitted	-	SBRT	I	21 months	KM
Hofheinz et al.	2016	NSCLC	31	PFS, OS	No	SUV, SURtc, Kslope	7.6	Fitted	PSF + TOF	CRT and/or surgery	T1-4N0-3M0	-	HR
Horne et al.	2014	NSCLC	95	LC, PFS, OS	Yes	SUVmax	5.0	Literature	-	SBRT	IA-IB	16.33 months	HR
Huyn et al.	2015	NSCLC	161	DFS, OS	Yes	SUVmax, MTV	14.0	Fitted	OSEM	Surgery +/− CT and/or RT	IIIA-N2	20 months	HR
Imamura et al.	2011	NSCLC	62	OS, PFS	No	SUVmax	6.0	Median	3D-RAMLA	CT or CRT	IIB-IV	464 days	KM
Jiang et al.	2018	NSCLC	151	OS	No	SUVmax	13.8	Median	-	CRT, RT, CT	I-IV	10 years	HR
Kohutek et al.	2015	NSCLC	211	OS	Yes	SUVmax, GTV	3.0	Fitted	OSEM	SBRT	T1-2N0M0	25.2 months	HR
Lee et al.	2012	NSCLC	205	OS	Yes	SUVmax	13.0	Fitted	Iterative	Neoadj. CRT, + surgery	IIIA	1.6 years	HR
Nair et al.	2014	NSCLC	163	PFS, OS, LRFS, DMFS	Yes	SUVmax	7.0	Median	-	RT, SBRT	T1-2N0M0	16 months	KM
*Nawara* et al.	2012	NSCLC	91	OS	No	SUVmax, SUVmean	7.0	Median	Iterative	RT +/− induction CT	I-IIIB	-	KM
Pyka et al.	2015	NSCLC	45	DSS, OS	Yes	SUVmax, SUVmean, MTV, COV, entropy, coarseness, contrast, correlation	11.2 (OS), 12.3 (DSS)	Fitted	OSEM	SBRT	T1-2N0M0	21.4 months	KM
Sasaki et al.	2005	NSCLC	162	OS, DFS	Yes	SUVmax	5.0	Fitted	Iterative	Surgery +/− RT or RT/CRT	I-IIIB	17 months	HR
Shirai et al.	2017	NSCLC	45	LC, PFS, OS	No	SUVmax	5.5	Median	-	C-ion RT	I	28.9 months	KM
Sugawara et al.	1999	NSCLC	38	OS	No	SUVlean	8.72	Median	Hanning filter	Surgery, CRT	I-IV	26.5 months	KM
Takeda et al.	2011	NSCLC	95	LC	No	SUVmax	6.0	Fitted	DRAMA	SBRT	IA-IIIB	16 months	KM
Takeda et al.	2014	NSCLC	152	OS, DFS. LC, RC, DMC, CSS	Yes	SUVmax	3.35 (LC), 3.64 (RC), 2.47 (DMC, DFS), 2.55 (CSS, OS)	Fitted	RAMLA	SBRT	T1-2N0M0	25.3 months	HR
Takeda et al.	2017	NSCLC	26	LC, PFS, OS	No	SUV, MTV, TLG, entropy, dissimilarity, HILAE, zone percentage	8.18	Median	OSEM	SBRT	T1-2N0M0	36 months	KM
Ulger et al.	2014	NSCLC	103	OS, RFS, DFS	Yes	SUVmax	10.7	Median	-	3D-CRT	IIIA-IIIB	22.63 months	HR
Vansteenkiste et al.	1999	NSCLC	125	OS	Yes	SUVmax	7.0	Fitted	-	Surgery +/− induction CT, RT +/− induction CT	I-IIIB	19 months (mean)	HR
*Vesselle* et al.	2007	NSCLC	208	OS, DFS	Yes	SUVmax	7.0	Fitted	Hanning filter	Surgery +/− neoadj. Or adjuvant therapy	I-IV	37 months	HR
Vu et al.	2013	NSCLC	50	OS, RFS	No	SUVmax, TLG, MTV	6.43	Median	-	SBRT	I	25.1 months	HR
*Xiang* et al.	2012	NSCLC	84	LRFS, DMFS, PFS, OS	Yes	SUVmax	14.2	Median	-	High dose proton + CT	III	19.2 months	HR
Yilmaz et al.	2018	NSCLC	67	PFS, OS	Yes	SUVmax	15.0	Median	-	CRT	III	20.7 months	HR

* Performed multivariate analysis (MVA) in regards to the endpoints analyzed in this study: DFS and OS. Studies listed with author in *italic* are performed as prospective studies. Other studies are retrospective studies. NSCLC: non-small cell lung cancer, OS: overall survival, SUV: standardized uptake value, SUR: standardized uptake ratio, RT: radiotherapy, CRT: chemoradiotherapy, HR: hazard ratio, LC: local control, AID: average iodine density, SBRT: stereotactic body radiation therapy, DSS: disease-specific survival, OSEM: ordered-subset expectation maximum, KM: Kaplan-Meier, MTV: metabolic tumor volume, DFS: disease free survival, PFS: progression free survival, TLG: total lesion glycolysis, CT: chemotherapy, RFS: recurrence-free survival, CSS: cause-specific survival, RR: regional relapse, LR: local relapse, DM: distant metastasis, LFF: local failure free rate, PSF: point spread function, TOF: time of flight, RAMLA: row-action maximum-likelihood algorithm, GTV: gross tumor volume, LRFS: local recurrence-free survival, DMFS: distant metastasis free survival, COV: coefficient of variation, DRAMA: dynamic row-action expectation maximization algorithm, RC: regional control, DMC: distant metastasis control, HILAE: high-intensity large-area emphasis, and 3D-CRT: three-dimensional conformal radiotherapy.

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
