# Peer review of "Multiple Testing, Cut-Point Optimization, and Signs of Publication Bias in Prognostic FDG–PET Imaging Studies of Head and Neck and Lung Cancer: A Review and Meta-Analysis"

_diagnostics, 2020, doi:10.3390/diagnostics10121030_

Round 1
Reviewer 1 Report
Extensive and well designed review paper. As the authors describes this field suffers from many misleading inforamtion. Please provide figure 4 in form with visible plots. I would recommend additional analyisis focusing on study comparison based on the tumor grade and FDG uptake relation analysis.
Author Response
Figure 4 has been updated in order to make the plots more visible.
Comparison based on tumor grade and FDG uptake could be an interesting analysis to make, however, these data are not available from the included studies.
Reviewer 2 Report
Dear authors,
I thank you for your very interesting manuscript. Please find my comments here under.
Line 42-43-44: Several studies have investigated the prognostic value of FDG PET, and dose escalation to PET positive areas within the tumor is one of the potential strategies for increasing effect of radiotherapy : please mark some references
Line 226 Figure 4: plots A & B in figure 4 are not completed (Bubbles missing)
Author Response
References has been added (line 42-44).
Figure 4 has been updated, and the bobbles are now in colours for a better visualisation.